# Barriers and Facilitators to HIV Treatment Adherence in Indonesia: Perspectives of People Living with HIV and HIV Service Providers

**DOI:** 10.3390/tropicalmed8030138

**Published:** 2023-02-24

**Authors:** Bona S. H. Hutahaean, Sarah E. Stutterheim, Kai J. Jonas

**Affiliations:** 1Department of Work and Social Psychology, Maastricht University, 6200 MD Maastricht, The Netherlands; 2Department of Clinical Psychology, Universitas Indonesia, Depok 16424, Indonesia; 3Department of Health Promotion & Care and Public Health Research Institute, Maastricht University, 6220 MD Maastricht, The Netherlands

**Keywords:** HIV, antiretroviral (ARV), antiretroviral treatment (ART), adherence, socioecological approach, Indonesia

## Abstract

HIV treatment adherence in Indonesia is a major challenge. Although previous studies have demonstrated several barriers and facilitators to adherence, studies providing a comprehensive analysis from both PLHIV and HIV service providers’ perspectives are limited, especially in Indonesia. In this qualitative study with 30 people living with HIV on treatment (PLHIV-OT) and 20 HIV service providers (HSPs), we explored, via online interviews, the barriers and facilitators to antiretroviral therapy (ART) adherence using a socioecological approach. Both PLHIV-OT and HSPs reported stigma as a major barrier at each socioecological level, including public stigma at the societal level, stigma in healthcare settings, and self-stigma at the intrapersonal level. Stigma reduction must therefore be prioritized. PLHIV-OT and HSPs also reported support from significant others and HSPs as the foremost facilitators to ART adherence. The enablement of support networks is thus an important key to improved ART adherence. Overall, the societal level and health system barriers to ART adherence should be addressed in order to remove barriers and enhance the facilitators at the subordinate socioecological levels.

## 1. Introduction

Indonesia initiated the use of antiretroviral therapy (ART) for people living with HIV (PLHIV) in the late 1990s. HIV is now considered a manageable chronic illness as long as PLHIV adhere to treatment [1]. However, ART adherence among PLHIV in Indonesia is low [2]. In terms of the 95-95-95 targets, the UNAIDS data for 2021 indicated that, of the estimated 610.000 PLHIV in Indonesia, only 66% are aware of their HIV status, 26% on ART, and no 2021 data was available on viral suppression [3], but earlier data also showed very poor rates of viral suppression [4,5]. Clearly, Indonesia, as a whole, does not meet the 95-95-95 targets [6] despite efforts, such as the 2012 continuum of care government initiative (Layanan Komprehensif Berkesinambungan) [7] and regional activities, such as the governor of Jakarta having signed the Paris Declaration in 2015 [8]. A number of reasons have been brought forward, such as poor retention in care [2], but also stigma and discrimination in society, as well as criminalization of key populations (e.g., men who have sex with men {MSM}, trans women), which creates barriers for accessing HIV services [9,10,11,12,13].

Adherence to ART is challenging, mainly because PLHIV are required to have 95% or near-perfect adherence to achieve viral suppression [14]. In Indonesia, of the 21.347 individuals reported to be on treatment, only 55.7% had good adherence to ART [15,16]. Poor adherence can result in drug resistance, greater risk of onward HIV transmission, and health deterioration leading to AIDS-related diseases or death [16,17,18].

Good adherence can lead to viral suppression, and thus prevent onward transmission [1,18,19,20,21]. An undetectable viral load makes HIV untransmittable [19,22,23]. As such, good ART adherence benefits not only the health and well-being of PLHIV; it also supports HIV prevention. This is termed Treatment-as-Prevention (TasP) [19].

Consistent ART adherence has many advantages, but it is not easy. There are many individual and environmental factors that can impede ART adherence. To effectively support treatment adherence in countries like Indonesia, we need a comprehensive understanding of both that which hinders and that which enables ART adherence. For this reason, we set out to explore the multiplicity of barriers and facilitators to ART adherence among PLHIV in Indonesia using a socioecological approach as our framework.

The socioecological approach argues that behavior is impacted by societal, organizational, community, interpersonal, and intrapersonal factors [24,25,26]. Societal level influences include values and beliefs in the society, as well as legislation and policy. Organizational level influences, such as health systems, shape behavior indirectly through, for example, organizational policies, protocols, and bureaucratic processes. The interpersonal level entails the influence of direct interactions between individuals and others, such as friends, family, support groups, and individual HIV service providers. Lastly, the intrapersonal level reflects internal influences on an individual’s nature in ways relating to their attitudes, beliefs, and knowledge [24].

Clearly, given the complexity of behavior and its influences, it is relevant to explore barriers and facilitators to ART adherence across socioecological levels. In addition, to gain a full understanding of the barriers and facilitators to ART adherence, it is also important to explore this from multiple perspectives. For this reason, we investigated barriers and facilitators to ART adherence in Indonesia from both the perspective of PLHIV on treatment (PLHIV-OT) and from the perspective of HIV service providers (HSPs), something that has not previously been done in the Indonesian context. In fact, to date, most studies investigating ART adherence in Indonesia focused only on one perspective, either the perspective of PLHIV or the perspective of HIV service providers [10,21,27,28,29,30,31,32,33,34]. We believe that understanding the barriers and facilitators from multiple perspectives is essential to effective interventions to improve ART adherence in Indonesia.

## 2. Materials and Methods

### 2.1. Study Design and Context

We conducted a qualitative study with semi-structured interviews among PLHIV-OT and HSPs to investigate barriers and facilitators of ART adherence. Data were collected in the greater Jakarta area, which covered five main areas of Jakarta as the capital (Central Jakarta, South Jakarta, West Jakarta, North Jakarta, East Jakarta) and the metro Jakarta area, namely Bogor, Depok, Tangerang, and Bekasi, abbreviated as Jabodetabek. Jakarta is characterized by a highly diverse population as well as a high proportion of growth every year due to migration from other provinces [35,36].

Ethical approval was provided by the Ethics Review Committee at Maastricht University’s Faculty of Psychology and Neuroscience (reference number: 188_11_02_2018_S17) and by the Health Research Ethics Committee at the National Institute of Health Research and Development in Indonesia (Badan Penelitian dan Pengembangan Kesehatan; No: LB.02.01/2/KE.139/2020). Research permits were provided by Dinas Penanaman Modal dan Pelayanan Terpadu Satu Pintu, Pemerintah Provinsi Daerah Khusus Ibukota Jakarta (No: 60/AF.1/1/-1.862.9/e/2020) and Dinas Kesehatan Pemerintah Provinsi DKI Jakarta (No: 2271/-1.779.3).

### 2.2. Sampling and Recruitment

We recruited a convenience sample of 50 participants for semi-structured interviews. Thirty were PLHIV on ARV treatment (PLHIV-OT) and twenty were HIV service providers (HSPs) including physicians, nurse practitioners, psychologists, and treatment companions or counselors working voluntarily at NGOs. Inclusion criteria for PLHIV-OT were: (1) having received an HIV diagnosis; (2) being on ART for at least one year; and (3) being willing and able to provide informed consent. Inclusion criteria for HSPs were: (1) working in HIV care for at least one year in a community health center, public hospital, or private clinic; (2) being willing and able to give informed consent.

Recruitment occurred initially purposively at community health centers (*Puskesmas*), public hospitals, and private HIV clinics, and then through snowball sampling from participants who had completed an interview. To recruit, we first sent documentation (i.e., research permits, ethical approval, and recruitment posters) to the targeted health centers, who put us in contact with eligible HIV service providers (e.g., physicians, nurse practitioners, treatment companions, and counselors) and PLHIV-OT. Once connected with potential participants, we explained the purpose and procedure for the study, the potential to withdraw at any time, and the fact that all data would be handled confidentially.

### 2.3. Participant Characteristics

The mean age of PLHIV-OT was 38.7 years with the average of 6.7 years for people on ARV treatment. Among them, more than half were cisgender men and self-identified as straight. The HIV service providers had a mean age of 34.7 years while more than half of them had between 7 to 14 years of professional experience in HIV care. In terms of formal educational attainment, 35% of the HSPs had vocational training and 25% had a bachelor’s degree, with only one of them never received any HIV-related formal training. Additional details are depicted in Table 1.

### 2.4. Data Collection

Interview guides were first developed in English based on the socioecological approach [24,25]. For PLHIV-OT, the main topics were participants’ experiences with an HIV diagnosis, and the barriers and facilitators to ART adherence at each socioecological level. For HSPs, the main topics were their views on their experiences caring for PLHIV, including challenges, and their views on barriers and facilitators to ART adherence.

The interview guides were then pre-tested with two English-speaking professionals working in HIV. After the pre-test, adjustments were made based on the feedback provided. Then, we translated the interview guides into Bahasa Indonesia and pre-tested them with two Indonesian colleagues, both of whom were HIV counselors, one of whom was also a researcher.

Single, face-to-face, semi-structured online interviews of approximately one hour were conducted between March and June of 2020 by the first author with assistance from four female postgraduate students who had extensive experience conducting qualitative research. Interviews were conducted in Bahasa Indonesia at a location chosen by the participants, usually the participant’s house or bedroom for PLHIV-OT or at the HIV clinic for the HSPs. No other people were present at the time of the interview. Interviews were preceded by informed consent, guided by the semi-structured interview guide with follow-up probes, and followed by a short survey measuring demographic and HIV-related or occupational characteristics for PLHIV-OT and HSPs, respectively.

Data collection coincided with the beginning of the worldwide COVID-19 outbreak. As a result, Jakarta and the metro Jakarta area imposed a large-scale social restriction policy (*Pembatasan Sosial Berskala Besar* [PSBB]) in March of 2020. The new measures limited social contact, and the interviews thus needed to take place online or by telephone. We utilized online mobile applications such as WhatsApp audio or video call, or online chat and video telephony software platforms such as Zoom to avoid direct contact with participants and ensure their safety from any COVID-19 risks. The participants who were willing to be involved in this study could choose the online platform that was most suitable for them. All participants agreed to their interviews to be recorded with a digital voice recorder. Video was not recorded.

### 2.5. Data Analyses

All of the interview recordings were transcribed verbatim in Bahasa Indonesia. Subsequently, all of the HSP transcripts and two of the PLHIV-OT transcripts were translated into English by a professional translator. The first author and the translator also reviewed the translated transcripts to ensure that the translations adequately reflected the original text. All authors were involved in the coding of the English transcripts and the first author coded the rest of the transcripts. We coded all of the transcripts using Atlas Ti V.8.4.5.

Thematic analyses was employed to identify, analyze, and report the recurring pattern of themes within the data [37]. We followed all six stages of thematic analyses [37]. First, we familiarized ourselves with the data by transcribing, reading, and making notes of the initial codes. Second, after the initial codes emerged, we collated the relevant data with each code. Third, we organized the codes into potential themes and merged the relevant data to each potential theme. Fourth, we reviewed the themes, resulting in an analysis thematic ‘map’. Fifth, we defined and gave names to each theme. Sixth, we generated the report of the analysis for this article. Selected quotes were translated to English and reviewed for the originality of meaning by the first author and the translator, who is fully bilingual in English and Bahasa Indonesia. An example of our theme development is provided in Table 2.

## 3. Results

Various barriers and facilitators to ART adherence emerged, some cross-cutting and some particular to one or more socioecological levels. Stigma was the main barrier appearing at every level. It manifested as public stigma at the societal level, stigma in healthcare settings at the health system level, stigma toward PLHIV-OT from significant others at the interpersonal level, and self-stigma at the intrapersonal level. Below, we outline the barriers and facilitators from the perspectives of both PLHIV-OT and HSPs at each socioecological level and describe how barriers and facilitators were interrelated.

### 3.1. Societal Barriers and Facilitators

Public stigma and discrimination in society colored adherence to ART. PLHIV-OT stated that stigma and discrimination were the major barriers that they had to face. They reported that society has an insufficient and incorrect understanding about HIV and PLHIV, which results in substantial stigma and discriminatory attitudes against PLHIV.

“*The challenge is from the society. There are still many people who look down on PLHIV, many people who stay away when they hear (HIV), many people still discriminate… We (PLHIV-OT) feel that we are just the same like everybody else, but they (society) still judge us.*” (PLHIV-OT, 31, public hospital)

“*Well, the things that prevent them (PLHIV-OT) from complying is their own fears about what they’re going to deal with, including stigma and discrimination from the society in general.*”(Psychologist, 26, community health center)

Stigma and discrimination in society created fear of being ostracized among PLHIV-OT, resulting in the non-disclosure of HIV status or being secretive about ART. However, the collectivistic culture in Indonesia makes it difficult as people in one’s neighborhood are likely to question why someone is secretive. This situation makes PLHIV-OT uncomfortable about doing anything related to HIV treatment (e.g., taking ARV medication, going to the clinic) around the neighborhood. They thus prefer to go to a clinic much further away, which leads to downstream hurdles that can potentially decrease their adherence. Moreover, PLHIV-OT who had already made an effort to adhere to ARV to reach viral suppression felt that their effort was unavailing when they were still being stigmatized and discriminated against.

“*…One of the challenges in undergoing the (ARV) treatment is the society. They still look us (PLHIV) one-sided or avoid us while we try to interact with them. Why is that a challenge? We (PLHIV-OT) take the meds, we cannot transmit the virus to others, but they (society) are afraid to shake our hands, let alone to have a conversation with us…*”(PLHIV-OT, 31, public hospital)

“*These patients’ adherence can be influenced from the stigma in the society. All of this time, the neighborhood might not notice if this patient doesn’t show any symptoms. But if he always goes to the hospital with his family for a routine check, there must be a question from the neighborhood: “Why does he always go to the hospital?*”(GP, 52, public hospital)

“*Stigma from the society since they are lacking of understanding about HIV population. This stigma can weaken PLHIV (adherence), lower their motivation, and make them slide back to where they were before*.”(Counselor, 38, public hospital)

The substantial stigma and discrimination in society also creates fear of being condemned, which affected PLHIV-OT’s willingness to continue ART, disclose their HIV status, or just be comfortable living with HIV. As a hope voiced by many PLHIV-OT and HSPs, less stigma and discrimination in society could contribute to better adherence. Moreover, less stigma and discrimination could contribute to support almost all of the facilitators at the subordinate levels, and certainly could help to overcome most of the barriers.

### 3.2. Health System Barriers and Facilitators

The health system barriers to adherence were stigma in healthcare settings, insufficient healthcare system coverage, complicated bureaucracy, and problems with ART access. Both PLHIV-OT and HSPs indicated that NGOs and HSPs have important roles in the provision of better healthcare coverage and easy access to ARV as the facilitator to adherence.

#### 3.2.1. Stigma in Healthcare Settings

PLHIV-OT and HSPs reported that there were many stigma-free and discrimination-free HIV clinics. However, stigma and discrimination at certain HIV clinics was still reported.

“*…I think some of them (HSP) still have stigma against PLHIV that it can make patients feel uncomfortable coming to the care center.*”(PLHIV-OT, 38, private clinic)

“*All I can think about is the healthcare providers who still have stigma against PLHIV. We have to be able to create a comfortable environment for PLHIV so that they’d feel welcome. If we fail to do that, then they will move somewhere else or find another center or hospital to get treatment, or worse, they completely stop the medication out of fear of being treated in the same manner as at the previous clinic they went to.*”(Nurse, 38, public hospital)

Even though many HSPs are knowledgeable about HIV and know standard procedures for interacting with PLHIV in the clinic, stigmatizing and discriminating attitudes and procedures in the health system were reported. These directly and indirectly affect adherence, particularly for men who have sex with men (MSM) and trans women. For example, trans women reported being called on by HSPs with their birth name listed on their identity card rather than their acquired name (i.e., deadnaming), and MSM who experienced stigmatization in clinics opted not to return and thus could not access ARV.

“*The transgender patients felt that they already put such effort to dress up like a lady so that people should address them as such. It’s a discomfort for them, but we have rigid system. If system allowed some changes to the rule, then they could add ‘alias’ next to birth name on ID card so that there’s option how or what to address a person*.”(Psychologist, 31, community health center)

“*For example, there was an experience of my colleague who took my shift, I didn’t really understand what happened, maybe she mistakenly said something that was considered as stigma for those MSM patients. They (MSM patients) didn’t want to come (to the clinic) again.*”(GP, 31, community health center)

#### 3.2.2. Healthcare System Coverage

The HIV healthcare system in Indonesia has improved over the years, but our participants still hoped that the government would provide more solutions to overcome the insufficient universal coverage in HIV care. ARV medication is provided for free by the government, but PLHIV who do not have a social security card (BPJS Kesehatan) still need to pay an administration fee of around IDR 15.000 (+/− USD 1) when they come to the HIV clinic in the community health center or public hospital. This amount is considered high, and thus is certainly a barrier, especially for PLHIV-OT who live in remote areas and have financial difficulties, considering Indonesia’s national income per capita in 2019 was around IDR 161.917 (+/− USD 11) per day, while the average national expenditure per capita was around IDR 40.195 (+/− USD 3) per day [38]. As a consequence, PLHIV-OT preferred to spend their money on groceries or basic needs rather than spending it on administration fees or transportation costs to get to their HIV clinic.

“*They (HIV patients) only pay retribution when they come into the clinic to register. It’s quite cheap—only IDR 15.000. However, for some people in Indonesia, IDR 15.000 is a lot of money to spend, in addition to transport cost of going back and forth from clinic to home.*”(Counselor, 38, community health center)

“*…It is really burdening for us (PLHIV-OT) when we have to pay for the CD4 and VL tests. Not everyone has enough money or BPJS…*”(PLHIV-OT, 37, public hospital)

In addition, those without social security have to pay for CD4 and viral load (VL) tests, which is often beyond their means. Even though there were some externally funded programs by the Indonesian AIDS Commission and other NGOs that provide free post-ARV tests, only a few PLHIV-OT could benefit from these programs. Moreover, the information available about these programs was not shared widely or came too late for them to make use of free tests.

“*…Indeed, the VL test is important, but I had an experience when they informed me about the free VL test two days before the event! It was definitely hard for the patients to get a permission from their daily job. If they want to make such an event, it will be better if they announce it weeks before the event…*”(PLHIV-OT, 26, community health center)

#### 3.2.3. Bureaucracy

PLHIV-OT with more income reported more choices for accessing ART, including private clinics which charged more in terms of administration fees and other laboratory tests, but the bureaucracy was less complicated compared to HIV clinics in public hospitals or community health centers. Meanwhile, PLHIV-OT who accessed ART in public hospitals or community health centers had to deal with more complicated bureaucracy as a tradeoff for a more affordable service.

“*First, most hospitals still require PLHIV patients to pay for registration fee. We need to waive the registration fee as most PLHIV are poverty-stricken. Second, we need to cut down bureaucracy and find a simple way to let patients get medication in any community health center or hospitals they want. We need to integrate the reporting and filing system so that patients can just show their registration serial number to get access to medication everywhere…*”(Counselor, 38, public hospital)

Unsurprisingly, HSPs reported that having less bureaucracy by cutting certain procedures could certainly facilitate adherence. One example would be allowing trans women or anyone without an ID card to access ART without their ID card. Even though the regulations state that patients always need to show their ID card, in order to access ARV, some community health centers have chosen to not comply with this regulation so that trans women in particular can access ARV without having to show their ID card. Participants also reported that some HIV clinics, particularly private clinics, already tackled bureaucratic barriers and provided faster access to ART and less queueing. This would also help to overcome the stigma in healthcare settings that we mentioned earlier.

“*…we (HIV service providers) also try to simplify the procedure. We have administration team who handle the (patient’s) data in our clinic. I usually give the administration staff number to the patients, so that they don’t have to queue every time they come to the clinic. These patients just have to make an agreement with my friend (the administration staff), and then he will register the patients. So, they don’t need to queue…*”(GP, 31, community health center)

“*There are many community health centers running different procedures with ours. We have a more flexible procedure because we think of what’s best for patients’ health. We bend a couple of regulation here and there such as we allow patients to come for VCT and ARV at our clinic without ID card because we know a lot of transgender community don’t have it*.”(Counselor, 38, community health center)

#### 3.2.4. ART Access

Faster access to ART as a result of less bureaucracy would be more complete with easy access to ART, which could be implemented in three ways. First, based on the data, it is advisable to implement a one-stop service where PLHIV-OT could come to the clinic, register, consult with the doctor, do tests if necessary, and acquire ARV at the same time and in the same place. This way, PLHIV-OT could access the medication with minimal effort by picking up their ARV medication easily in the clinic instead of the pharmacy. This is not only advantageous because it is more efficient; PLHIV-OT also reported feeling more comfortable going to just one place, which reduces the chances of running into people they know. Evidently, internalized and anticipated stigma as barriers to ART can be minimized by enabling one-stop access to ARV medication in the clinic.

“*…I’ve heard there is this Puskesmas which has a very simple system. It’s really good. Starting from the registration, then the weight and blood pressure measurement, then directly to the consultation with the doctor, and the doctor gives the medication directly to the patient. All at one place. So, they don’t have to pick up the medication at the pharmacy. That’s really comfortable.*”(PLHIV-OT, 50, community health center)

“*…At the time being, what is helpful is giving the medication at VCT clinic directly, not at the pharmacy counter … Most transgender patients, if not all, like to be called by their female’s name, not with their old male’s name on ID card. When they pick up medication at pharmacy, the counter will always use patient’s real name from ID card, and transgenders population don’t like that very much. So, they are very happy to pick up medication at our clinic because no one there is exposing their real male’s name*.”(Nurse, 27, community health center)

“*…One-door service! The medical devices should be provided more to be able to give ‘one-door service’. The patients just have to come to ‘one door’ when they want to do an X-ray, VL, CD4 test, or get a TB and STDs treatment. They can feel more comfortable and don’t have to run back and forth just to take those tests … Thank God we already have it all here in Puskesmas X*.”(GP, 38, community health center)

The second way in which access to ART can be facilitated at the health system level is by providing ARV supplies to PLHIV-OT for two or three months at once, instead of for only one month, so that the patients do not have to come to the clinic every month. This was really helpful for PLHIV-OT with financial constraints that limit their ability to travel back and forth to the clinic.

“*…Nowadays, when the healthcare providers see good progress from a patient, they give meds for 2-month supply. It helps patient’s adherence to the medication and avoid them coming every month…*.”(PLHIV-OT, 34, private clinic)

Third, easy access to ART can be implemented by allowing the patients with good adherence to have their ARV medication delivered using an online taxi bike application (ojek online). This ARV medication delivery service is an informal collaboration between HIV clinics and treatment companions from the NGOs (Pendamping Minum Obat; PMO). However, PLHIV still need to come and consult with an HIV physician one month after ARV medication delivery for a routine check. PLHIV-OT who have a good relationship with their health care provider can ask for their help for ARV medication delivery or ask the treatment companion to deliver it. Therefore, having a good client–provider relationship at the interpersonal level, which we will explain later, can support easier access to ART for better adherence.

“*…We make things easier too nowadays. NGO also steps up and helps deliver the medication to them. So, there’s no excuses to not taking it.*”(Counselor, 25, community health center)

“*We do have a lot of requests for delivery during this corona pandemic, but only with one condition: compliance to taking the medication regularly … For example, if we see that the patient has a good viral load, then we will hire a delivery service to send the medication for him. But if we see that his last VL result is bad, and he is always a week or 2 weeks late to pick up medication, then we ask the patient to come to the clinic for re-consultation.*”(Nurse, 27, community health center)

Despite increased opportunities for easy access to ART, PLHIV-OT who experienced the ARV stock crisis in 2015 or 2017, and when the COVID-19 outbreak happened in early 2020, voiced concerns. Insufficient ARV stock made it difficult for them to access ART for a one-month supply as usual. Instead, physicians prescribed them for only one or two weeks with different types or amounts of ARV, which led to patients having to adjust to a new regimen, including side effects.

“*…That’s right, I got more than one pills (when there was ARV stock crisis). I was even a bit worried whether there would be another side effects or not…*”(PLHIV-OT, 28, public hospital)

It was very inefficient and time-consuming, especially for PLHIV-OT who had inflexible schedules but had to come to the clinic almost every week, which brought about extra transportation costs. PLHIV-OT with financial constraints often waited until the ARV stock was back to normal again before reinitiating ART. This created an adherence gap. Although the ARV stock crisis is not currently an ongoing barrier, it still can create lingering anxieties for PLHIV-OT.

“*…I’m grateful for the free ARV, but I think the government needs to rethink its management plan on ARV national stock supply as these meds need to be taken for a lifetime. In addition, there’s been rare stock supply in region of Aceh, Padang, Ambon, Bali, and Manado. Most PLHIV there only get 14-day supply and not a full 1-month supply*.”(PLHIV-OT, 30, private clinic)

In conclusion, health system level barriers to ART adherence include stigma, insufficient universal coverage in HIV care, and complicated bureaucracy. Informal collaboration between HSPs and NGOs that enables ARV medication delivery, as well as reduced bureaucracy and easy one-stop access to ART, can serve to overcome some of the barriers to ART.

### 3.3. Interpersonal Barriers and Facilitators

Interpersonal level barriers were stigma from the PLHIV-OT’s significant others or their own HSP, which led to, among other things, poorer client–provider relationships and, consequently, lowered the adherence. PLHIV-OT reported support from significant others as a major facilitator and, surprisingly, HSPs mentioned that the use of fear appeals facilitates adherence.

#### 3.3.1. Significant Others

Significant others, such as friends, family members, and (ex)partners, played important roles both as barriers and as facilitators to ART adherence. As a barrier, significant others who stigmatized and discriminated against PLHIV by rejecting or excluding them (i.e., enacted stigma) directly and indirectly affected adherence. For example, PLHIV-OT who were ostracized by their family or friends or rejected by a partner or potential partners after disclosing their HIV status were at risk of discontinuing ART because they would potentially feel depressed and thus did not come to the clinic to pick up their ARV medication.

“*I once dated a guy and just being straight forward by telling him my (HIV) status. He said: “Maybe I’m not your match. I hope you’re healthy.” And then he ran away (laugh). I thought I might just tell him than regret it, right?*”(PLHIV-OT, 48, public hospital)

“*…We have several patients who were shunned by their family. They got so depressed and never picked up medication again. That’s how we lost contact with the patients. We don’t know where they are until today*.”(Counselor, 38, community health center)

Significant others who accepted the HIV status of their partners functioned as a facilitator to ART adherence. They could provide sufficient emotional and instrumental support, but this was reported to not be immediate. PLHIV-OT who disclosed their HIV status to friends and family often reported initial difficulties gaining acceptance from others, but then receiving support later, and this helped facilitate their adherence to ART. Specifically, they described emotional support, such as acceptance and listening to treatment challenges, and instrumental support, such as accompaniment to the HIV clinic or pharmacy. According to HSPs, PLHIV-OT receiving support, particularly from their family, had a better adherence than those who did not.

“*…The form of support is not only about reminding us to take the medicine on time, but also about accepting our condition, that we are different, we have a ‘special’ situation…*”(PLHIV-OT, 31, public hospital)

“*Family maybe is more important (than friends). Most patients are brought to the clinic by their family member, the closest family member … I think they can help reminding patient to take medication*.” (Nurse, 27, community health center)

#### 3.3.2. Client–Provider Relationships

Not surprisingly, given its presence at the societal and health system levels, stigma was also apparent in interactions between HSPs and PLHIV. A poor client–provider relationship, caused by patients who were not very communicative or stigmatizing HSPs, often led PLHIV-OT to discontinue consultation. Examples included HSPs who disrespectfully told a trans PLHIV that trans PLHIV almost always fail to adhere to ART and HSPs who expressed judgment about sex between men.

“*The doctor saw my ID card (listed as a male) and asked me why do I look like this (female appearance). He said that my chance (as a trans woman) of not complying to the treatment would be much greater throughout this lifetime treatment. He didn’t respect me at all. And I think he did this not only to the trans patients, but also to the MSM community.*”(PLHIV-OT, 26, community health center)

[Telling stories before moving to the current HIV clinic] “*I realize that we (patients) have to be honest to our providers during consultation. So, I told my doctor that I am gay. He instantly said: “Why are you gay? Aren’t you afraid of sins, being punished, this and that? You must repent from your sins!” I came to him not to listen to his religious lecture. If you are a healthcare provider, just do your job as a healthcare provider*.” (PLHIV-OT, 31, public hospital)

To overcome poor client–provider relationships, HSPs encouraged their patients to be more proactive in asking questions and discussing matters regarding their treatment. They observed that the PLHIV-OT who speak up, ask questions, or complain more tended to have better adherence.

“*…I noticed that the more they (PLHIV-OT) complain, the more adhere they are to the treatment. Because when they start to complain, they will text us through WhatsApp, asking lots of questions etc., and it’s easier for us to do a little bit of counseling through the text and always remind them to adhere.*”(GP, 38, community health center)

In other words, PLHIV-OT who felt comfortable being assertive showed that their providers facilitated a safe atmosphere for them to ask questions during clinic visits, by dedicating more time to patients, providing more information, and acting in a friendly manner. This safe atmosphere created a good client–provider relationship that motivated PLHIV-OT to interact more with their providers, come to the clinic, and adhere to treatment. This healthy relationship could also be a form of adherence support, which was a prominent facilitator mentioned by both PLHIV-OT and HSPs.

“*The truth is, when we’re building a good relationship with patients, they’ll be comfortable enough to talk about anything with us. We need to have that kind of relationship in order to keep this medication going. It’s a way to reach out to them and support them*.”(Counselor, 35, community health center)

Lastly, in addition to HSPs in the clinic, PLHIV-OT also needed to maintain a good relationship with other HSPs such as the NGOs that explicitly provided a “helping hand” to support adherence.

#### 3.3.3. The Role of NGOs and Use of Fear Appeals

To be able to receive additional support from outside the clinic, PLHIV have to disclose their HIV status to people they can trust. Unfortunately, not all PLHIV-OT were willing to disclose their HIV status to family or friends, mostly because they, justifiably, feared being stigmatized and discriminated against. In order to overcome this, PLHIV-OT could join peer support groups (Kelompok Dukungan Sebaya) which commonly exist under the umbrella of certain NGOs. These groups usually comprise PLHIV who do not know each other before joining the group but that become friends who share treatment experiences and support one another. Having this opportunity to experience affinity with other PLHIV can make patients feel more comfortable sharing their stories and may help them feel supported. Within the context of these support groups (but also beyond), other PLHIV’s success stories regarding ARV served as positive examples and were perceived to have an advantageous influence on adherence.

“…*these support group and HIV organization can also be a support system. Some patients don’t have any support at all; the family totally rejected them. In this case, the friends from NGO can be their support system - not in picking up their medication or anything, but in a sense of what support system really does*.”(GP, 36, community health center)

“*I believe I’m most comfortable with the testimonial from the people living with HIV. They shared not only on their life story, but also the side effects of the ARV treatment. The doctor can only give me the explanation about it, but not actually feeling what I’m going through as none of them are HIV positive.*”(PLHIV-OT, 35, private clinic)

“*…I think they (peer support group) are great. When someone with HIV feels the world is rejecting them, it gives them a place to go when they want information or simply just to have a sanctuary, or to be comfortable knowing that they are not the only one.*”(Psychologist, 26, community health center)

Nonetheless, HSPs reported that apprehensive stories or pictures, such as reminding a patient not to stop ARV by sending pictures of untreated PLHIV taken from the internet, could serve as “good” role models for the PLHIV-OT as it would keep them from discontinuing ART. HSPs felt that PLHIV-OT would be more likely to adhere if they were exposed to messages of how other PLHIV suffered after discontinuing treatment. These are termed, in psychology, fear appeals, and are known to be usually ineffective in the absence of self-efficacy and skills.

“*…I haven’t found out how to manage boredom. I am now back to taking the medicine not because I am obedient, but because I am afraid … Thankfully, my adherence is getting better these past two months because of fear, fear of death, fear of creating problems to other people…*”(PLHIV-OT, 45, public hospital)

“*…If we have a patient died, we share it on the group chat to encourage them not to forget to take medicine routinely … If they (HIV patients) don’t want to take it (ARV), we show them a picture of someone without taking medication. That usually does the trick. Thankfully, we have some that started taking it again yesterday*.” (Nurse, 28, community health center)(Nurse, 28, community health center)

#### 3.3.4. Influence of (Social) Media

The content of some social media accounts was also reported to be a barrier to ART adherence. Unreliable social media accounts (e.g., the anti-ARV Facebook group called Mahastar) provided incorrect information regarding HIV and ART. They suggested PLHIV stop ARV and use alternative medication as the way to cure HIV so that PLHIV would not be dependent on it and provided testimonies from PLHIV who apparently discontinued ART but were still healthy.

“*…There are so many anti-ARV movements in Facebook. That kind of wrong information will make them unstable in making decisions to start or even to undergo the (ARV) treatment*.”(GP, 38, community health center)

“*Nowadays, we can easily ‘ask’ Google about anything, even about the medicine to cure HIV, or herbal medicine. There were lots of unfortunate cases from our friends (PLHIV) who stopped taking ARV and decided to take the herbal medicine because they thought that it could cure them. In the end, they caught typhoid, got AIDS, and then regretted and realized that ARV was the only medicine that they needed, not the herbal medicine.*” (PLHIV-OT, 36, community health center)

“*It’s never proven successful clinically. I advise the patients to fight this notion on herbal medicine … So, we do have a lot of challenges, and one of them is the anti-ARV*.”(Counselor, 35, community health center)

Despite the incorrect information about HIV and ART on social media, there were still credible HIV-related social media accounts on Instagram, Twitter, or other platforms that could facilitate PLHIV-OT to gain more reliable information and potentially help them adhere to ART. For example, participants described PLHIV disclosing their HIV status on Instagram or YouTube, sharing their stories of accepting HIV, talking about being on ART, and showing that they could live healthy lives with HIV by taking ART. By viewing these inspiring stories, PLHIV-OT felt they had more knowledge about HIV and ART, and looked up to the living-proof examples of ART adherence, which increased their own adherence. As an additional form of support, HSPs shared examples of credible social media accounts with PLHIV.

“*I can ask them (social media accounts) about some things. For example, before I moved to Jakarta, I asked (certain social media) “Is there any HIV clinic in Jakarta that provides access to the medicine on weekends?” They gave me choices ‘here, here, here’ and so, I kept looking on the internet too, and it turns out that they (HIV clinics) have social media.*”(PLHIV-OT, 28, public hospital)

“*…Most PLHIV know these accounts and the accuracy of the information there, therefore it can help PLHIV to comply with ARV medication. These accounts give a clear idea of the risk if they don’t take the medication*.”(Nurse, 30, community health center)

Distributing correct information about HIV and ART through social media not only had a direct influence on the PLHIV-OTs’ adherence; it also indirectly increased awareness in society, offering increased opportunities for the provision of better support to PLHIV and hopefully decreasing stigma and discrimination.

“*…When the public has good education, they tend to be able to process new information logically. They will use critical thinking and they’ll want to recheck the information given from those platforms further. It’s absolutely beneficial having these platforms spreading news about it*.”(Psychologist, 31, community health center)

In sum, the interpersonal level barriers to ART adherence included poor client–provider relationships and incorrect information regarding HIV and ARV from unreliable social media accounts. Facilitators to ART adherence were often the opposite or the absence of the barriers and frequently took the form of support. Being proactive with HSPs was also a strong facilitator for improved client–provider relationships, and disclosure of HIV status, when safe, was claimed to be helpful in helping PLHIV gain more support outside the clinic. Internet or social media content also played an important role both as barriers and as facilitators, depending on the aim and content provided.

### 3.4. Intrapersonal Barriers and Facilitators

Intrapersonal level barriers and facilitators to ART adherence were the most mentioned by both groups of participants, starting from self-stigma, with barriers and facilitators starting from the moment they initiate treatment, but also occurring after years of treatment.

#### 3.4.1. Self-Stigma

PLHIV-OT reported that HIV is often thought of as a “cursed disease” and, not unsurprisingly or unjustifiably, they were scared that people would ostracize them. This internalized and anticipated stigma made them approach treatment in a complicated way that required efforts to hide their HIV status so that others would not know they had HIV. For instance, PLHIV-OT reported choosing clinics much further from their home and disposing of empty ARV boxes in public waste bins away from their home so that others would not find out they have HIV. Clearly, PLHIV-OT spend subsantial time and effort concealing their HIV treatment.

“*Every time I want to disclose or share my stories to others, I have an automatic answer in my head: “They will alienate me.” That’s why I choose to be secretive … I don’t want to be ostracized. If I disclose my (HIV) status, is it possible that people would accept me? I think it’s impossible*.”(PLHIV-OT, 46, public hospital)

“*…But not all patients want to get the treatment in their hometown, with many reasons, mostly because they feel ashamed.*.”(GP, 31, community health center)

“*Every month, I never throw the ARV box in the household waste because I am afraid that my family would find the box and search the info about the medicine*.”(PLHIV-OT, 33, public hospital)

Self-stigma in PLHIV was driven by perceived stigma in society, having observed stigma in healthcare settings, and previous experiences with enacted stigma. By internalizing negative beliefs about HIV and PLHIV, PLHIV-OT often alienated themselves from their surroundings and did not see that there might be people who would not reject them and were even willing to support them. Self-stigma thus precluded them from seeing possibilities to gain acceptance and support from others.

#### 3.4.2. Side Effects and Knowledge Gaps

Both PLHIV-OT and HSPs reported that challenges related to ART initiation were one of the major barriers to later ART adherence. Side effects (e.g., fever, nausea, vomiting, or feeling sluggish and tired), experienced by most PLHIV-OT, impacted their attitude towards ART adherence later.

“*A lot of patients complain about the horrible side effects they’ve been experiencing. There are some that cut the meds off completely, thus, lost to follow up due to the problematic side effects that they couldn’t endure…*”(PLHIV-OT, 39, community health center)

“*…Secondly, the side effects can stop their adherence. The patients who just start the treatment barely can stand the side effects; hence, it will make them stop to take the medication*.”(GP, 38, community health center)

HSPs believed that PLHIV-OT who had a better understanding of HIV and ART would also be more willing to continue ART despite initial side effects. PLHIV-OT commonly thought, upon receiving their HIV diagnosis, that they would have a short life and inevitably die. That changed after HSPs explained how ART could suppress their viral load and increase their life expectancy. Some PLHIV-OT then sought out more information on the internet or social media, and/or had more discussions with their HSP leading to better knowledge about HIV and ART.

“*…the person’s knowledge on HIV really affects his adherence to meds. Once he knows how to treat his illness, it’s easier for him to adhere to the meds given*.”(PLHIV-OT, 34, private clinic)

“*…By getting more information about HIV from social media, it empowers them more to stay healthy and therefore, adhering to the medication…*”(Counselor, 35, community health center)

#### 3.4.3. Reduced Motivation

Problems with adherence can also arise later in life, long after HIV diagnosis. PLHIV-OT who had been on ART for five years or more said the major barriers were a lack of motivation to continue treatment or simply feeling lazy, bored, or forgetful. In general, taking ART on a daily basis for the entire duration of one’s life was considered tedious.

“*I think it’s really human when we (PLHIV-OT) are bored of taking these meds every day. It’s the same feeling when we have with eating rice with the same side dish every day…*”(PLHIV-OT, 50, community health center)

“*I’m adhering to the meds, but still, I sometimes forget whether I have taken one in a day, so yes, I kind of skip a couple of times because I forget*.”(PLHIV-OT, 39, community health center)

According to HSPs, a lack of motivation, laziness, boredom, and forgetfulness might reflect mental health issues. HSPs claimed that many PLHIV develop depressive symptoms after HIV diagnosis and, correspondingly, ART initiation, but also much later in their care trajectories. It is possible that PLHIV-OT are also in denial about their HIV status, which leads to dissonance with ART adherence. Additional, ongoing substance use dependence can further impede ART adherence.

“*…I think they (PLHIV-OT) are prone to experience a psychological disorder and when it’s not properly treated, that will certainly affect how they’re going to conduct their daily routines. That’s why they need a well-balanced and stable mental health on day to day basis… When there’s hindrance psychologically to initiate or maintain medication, that can cause thought blocking. A negative behavior will create itself…*”(Psychologist, 30, community health center)

“*They (IDU) don’t think the way normal people do. They only think about how to get to the next drug or syringe. They only come (to the clinic) when they’re already in a bad condition. Even then, they will discontinue the ARV medication once they don’t feel the need of taking it again*.”(Counselor, 35, community health center)

#### 3.4.4. Lightening the Load with Euphemistic Terminology

Some PLHIV-OT reported unique ways to make ART adherence more bearable and less tedious. For example, some used euphemistic terminology by addressing ARV as “vitamins”, “supplements”, or “beauty pills” because ARV medication, like vitamins or supplements, is taken every day, and because ARV medication can make PLHIV-OT healthier and physically more aligned with beauty ideals. Clearly, the use of euphemistic terminology was employed to ‘lighten’ the load of daily ART adherence.

“*…We don’t want to think of it as a drug, but consider it a vitamin. If we take vitamins, we want to be healthy, right?*” (PLHIV-OT, 33, public hospital)

“*I’ve always told PLHIV that this drug is a ‘vitamin’. Drug or medication has such a bad stigma in their mind. By switching the term to ‘vitamin’, they don’t really think of it as an obligation, rather, as daily activity that they need to do every day…*.”(Nurse, 35, community health center)

#### 3.4.5. Meaning-Making through Goals and Spirituality

Having clear goals or intentions, such as wanting to get married and have children, seeking a promotion at work, or pursuing higher education, was also reported to facilitate ART adherence. Having goals demonstrated hope for the future, feeling motivated, and having reasons to live through ART adherence.

[Talking about the things that can help to adhere to ART] “*…My kids, my mom. My kids mostly. I want to be able to watch them grow up, finish college. That’s eveyone’s hope, I guess. I’ll use the rest of my time given by the highest power above to continue this therapy if that is what it takes to be with my kids.*”(PLHIV-OT, 39, community health center)

“*Like I’ve said before, the things that can help them (PHIV-OT) to comply is their desire to live, their desire to be productive, and their desire to live with their family. For example, a sex worker patient has a desire to get married, to have a new life. That’s really helpful for her to comply to ARV. Another example is from a housewife who has a child: She thinks that she has to stay healthy in order to work and raise her child. That’s also really helpful for her to stay on ARV…*”(GP, 38, community health center)

Having hope and a reason to live as the result of having goals was supported by other facilitators, namely religion and spirituality. Some PLHIV-OT believed that their HIV diagnosis provided them with a “second chance” in life, particularly for those who realized that their engagement with risky behaviors had led to their HIV diagnosis. These PLHIV-OT tended to view ART adherence as a necessary way to demonstrate their gratitude to God for their new ‘chance’ at a “second life”. PLHIV-OT set goals after realizing that they could live long and healthy lives with HIV, and they reported trying to live life to the fullest.

“*…In fact, I am increasingly convinced that this (ART) is proof of God’s love for me. God’s love is universal. I was given a second chance to continue living. So, I have to do my best (by adhering to the treatment).*”(PLHIV-OT, 33, public hospital)

“*I believe in God, and I believe God always ask us to cooperate. At least that’s what they always remind us about during the worship session. We cannot just ask, ask, ask for a healthy life from God, but we don’t try to live a healthy life*.”(PLHIV-OT, 37, public hospital)

#### 3.4.6. Fatalism

Although having goals and spirituality brought about more hope and motivation to adhere to ART for some PLHIV-OT, for others, a belief that all things in life, including having HIV, is predetermined and therefore inevitable, served as a barrier to ART adherence. HSPs reported that some PLHIV hold the fatalistic belief that their health destiny is in the hands of God and, because of that, there is no reason to take ARV medication.

“*…I heard couple of patients hopelessly said: “What will be, will be.” Some undisciplined patients also tend to use the same tone of language and adding: “We all eventually will die anyway. I know there’s a widely spread notion out there that there’s no hope for them. That’s why a lifetime commitment doesn’t register well in their brain, especially for the depressed patients. They tend to say: “Why am I still not cured? I’ve been taking this medication for ages.*”(Psychologist, 28, community health center)

Similarly, PLHIV-OT experiencing depression and the impacts of stigmatization felt that lifetime treatment was useless and made them feel even more hopeless, which impeded adherence.

“*…Once, I had a difficult patient who liked to throw cynical remarks at us by saying: “Death is on God’s hands…so relax…” Sometimes, that kind of remark hurts our feelings because it defies our continuous effort to keep them safe and alive… Compliance is the ultimate challenge*.”(Nurse, 27, community health center)

In conclusion, intrapersonal level barriers to adherence included self-stigma, side effects, knowledge gaps, reduced motivation, mental health difficulties, and fatalistic beliefs. These barriers could be overcome by improving knowledge about HIV and ARV, referring to ART euphemistically, setting goals or intentions, and making meaning through religion and spirituality which provides a sense of hope for PLHIV-OT.

## 4. Discussion

This study presents, from the perspectives of both PLHIV-OT and HSPs, facilitators and barriers to ART adherence in Indonesia across socioecological levels. The findings clearly show that stigma and discrimination are the main barriers to ART adherence at each socioecological level, and that reduced stigma and discrimination in the future can facilitate ART adherence. Furthermore, the role of treatment companions or buddies from NGOs can help to reduce barriers at the health system level, and peer support groups can facilitate PLHIV-OTs’ adherence at the interpersonal level. Additionally, intrapersonal level barriers such as side effects, knowledge gaps, reduced motivation, mental health difficulties, and fatalistic beliefs can be tackled by improving knowledge or lightening the ‘load’ of ART adherence by creating a sense of purpose through goal setting and spirituality.

Across the findings, participants reported perceiving substantial public stigma in society, institutional stigma in the health system, discrimination in contact with HSPs, enacted stigma from significant others, and anticipated and internalized stigma in PLHIV themselves. That HIV stigma is a major impediment to ART adherence is well established in the literature [9,13,39], as is the fact that HIV stigma in health care is highly prevalent and highly detrimental for PLHIV [40,41,42,43]. As such, our findings dovetail with previous studies reporting that stigma in any form hinders treatment adherence [3,11,27,30,31,32,42,44,45,46,47,48,49,50,51,52,53,54,55,56,57,58,59]. PLHIV also often experience intersectional stigma or stigmatization due to multiple identities within marginalized groups (e.g., gender or sexual orientation) [60,61].

Indeed, our study has shown that the many forms of stigma—perceived, enacted, anticipated, and internalized—serve as barriers to ART adherence and interact with one another [9,13,62]. Therefore, if we are to improve adherence, we must reduce HIV stigma and discrimination in Indonesia [48,60,63,64]. There are several ways to successfully reduce stigma [65]. Some include activating self-acceptance through empowerment and education to reduce self-stigma at the intrapersonal level, improving HIV and ART knowledge through peer support and training for HSPs at the interpersonal and health system level, and collaborating with community leaders (e.g., religious leaders) to implement a more humanistic approach towards PLHIV at the societal level [65,66]. The utilization of technology, such as the internet, social and mass media, in ways that provide more positive information, as well as articles reflecting a human perspective on HIV can bring about greater empathy in society and can actively help to alleviate public HIV stigma [41,67]. Additional efforts that seek to change beliefs, such as beliefs that HIV is highly contagious and a condemned disease, and breaking cultural taboos on HIV and sexuality, should also be embedded in stigma reduction interventions [43,65,68].

In addition, ascertaining that stigma is the foremost barrier to ART adherence, our findings have illuminated additional, sometimes temporary, barriers, such as the ARV stock crisis. During the initial outbreak of COVID-19 in Indonesia, significant delays in the ARV shipments from India to Indonesia occurred [69]. The pandemic had a substantial negative impact on PLHIV-OT who were prescribed different and often less ideal drug regimens. As a result, thousands of PLHIV-OT discontinued treatment [70]. Even though the pandemic is now well handled, our participants strongly voiced the need for better policies to ensure that ARV stock remains stable even when unforeseeable circumstances arise.

In terms of facilitators to ART adherence, our findings showed that support is a prominent facilitator to ART adherence at the interpersonal level. Previous studies have also reported that receiving sufficient support can, and does, facilitate adherence and can act as a buffer against psychological distress and other mental health issues (e.g., depression, anxiety) in PLHIV [11,12,21,71,72]. However, in order to receive support, disclosure of HIV status is often necessary [56]. This can be difficult, and potentially unsafe, for PLHIV-OT who anticipate stigma and discrimination from others. Several studies reported that PLHIV-OT who want to disclose their HIV status should receive counseling where they can weigh the risks and benefit of disclosure, and mentally prepare for a wide range of reactions to their status disclosure [73]. Furthermore, counseling can improve their ability to provide accurate information about HIV and ARV to others after disclosing their HIV status [12]. It is important to state that HSPs should not immediately think that a disclosure of HIV status is the best (and only) choice for PLHIV-OT [74,75].

Our interpersonal level findings on facilitators to ART adherence indicated that the HSPs in this study felt that PLHIV-OT could experience support and increased motivation through the provision of fear appeals. The WHO HIV treatment guidelines state that this is unethical [76], and a previous review has clearly demonstrated that it is generally ineffective in the absence of self-efficacy and skills [77]. We strongly advise against the use of fear appeals.

Overall, this study provides a comprehensive overview and interpretation of barriers and facilitators to ART adherence in Indonesia. We believe this study adds important theoretical and practical knowledge to previous studies on ART adherence, particularly in resource-limited settings such as Indonesia. We also hope that our findings can provide input and impetus to adjust policies on ART supply, stock, and treatment procedures, as well as non-discrimination of PLHIV. We further suggest that both PLHIV-OT and HSPs work on enhancing the facilitators described in our study and reducing the barriers. For example, HSPs can offer suggestions to PLHIV-OT, even practical and seemingly minor coping strategies such as speaking about ARV medication euphemistically. They can further help PLHIV-OT to set clear goals to increase hope and motivation to adhere to ART. PLHIV-OT can proactively seek out HIV and ARV knowledge through reliable sources to have a good understanding on why ART adherence is important. In addition, both PLHIV-OT and HSPs should work on establishing and maintaining a good relationship with one another.

Our findings also have additional implications for clinical practice. First, we strongly recommend that HCPs provide a reliable and easy to access, one-stop treatment service for PLHIV-OT with consistent long-term follow-up. This requires significant effort to address barriers to ART adherence that can be overcome (e.g., lack of discipline or forgetting to take the ARV, lack of knowledge and awareness). Additionally, it is advisable to create optimal conditions for the establishment and maintenance of a solid support system for PLHIV-OT either from their significant others or a peer support group. Furthermore, it is important that HSPs continuously improve their awareness and knowledge of HIV care and counseling, potentially through training sessions and workshops, so that they can provide the most optimal service.

Our study has several strengths. One is our comprehensive approach where we investigated barriers and facilitators to ART adherence from both the perspectives of PLHIV-OT and the perspectives of HSPs. This allowed us to effectively identify the overarching and unifying theme of stigma as the major barrier to ART adherence. We believe that our triangulation of data across PLHIV-OT and HSPs is a major strength. Both perspectives are extremely crucial for successful ART adherence. Another strength is our sample size. We successfully recruited and included a robust number of participants (n = 50), which provided the basis for comprehensive results, and offered insights into unexpected barriers and facilitators such as social media (both positive and negative) and spirituality (again with both facilitating and inhibiting effects).

In spite of these strengths, our study also has limitations. First, sampling occurred mostly in Jabodetabek, which is located in the metro Jakarta area, or inside and around the capital of Indonesia. We did not sample in other provinces. However, based on our noting of places of origin, our participants did represent the diversity of people in the Indonesian population relatively well. However, it would be advisable to conduct similar work in other provinces, particularly those with high HIV prevalence and/or low levels of adherence. A second limitation is the fact that our participants are a convenience sample. We recruited purposively based on inclusion and exclusion criteria and followed this up with snowball sampling. The risk of this approach is that new participants were referred by participants with similar opinions. However, it is important to note that generalizability is not the goal of qualitative research [78] and that our 50 participants were a diverse group in terms of age, gender, educational attainment, and professional and HIV-related characteristics. Another possible limitation was that PLHIV-OT, by virtue of them being on treatment, have experience with treatment and may have more favorable views on ART. It is therefore important to also explore barriers and facilitators to ART initiation.

## 5. Conclusions

Stigma is the foremost barrier to ART adherence across every socioecological level, and it converges with a number of other barriers starting from treatment initiation until after years of treatment. These barriers can be overcome by complementary facilitating factors that ameliorate the barriers at each socioecological level, and, for this to be effective, collaboration between PLHIV-OT, their significant others, HSPs, and NGOs is necessary. Overall, it is crucial to intervene by starting from the most abstract societal level, so that facilitators can be bolstered and barriers can be overcome at the subordinate levels.

## Figures and Tables

**Table 1 tropicalmed-08-00138-t001:** Participant characteristics.

Characteristics	PLHIV-OT (n = 30)		HIV Service Providers (n = 20)	
	**N**	**%**	**N**	**%**
** *Age* **				
25–35	12	40	11	55
36–45	11	37	8	40
46–59	7	23	1	5
** *Gender identity* **				
Cis male	17	57	10	50
Cis female	10	33	10	50
Trans woman	3	10	na	na
** *Educational background* **				
Below High School	2	7	-	-
High School	15	50	4	20
Vocational School	7	23	7	35
Bachelor	6	20	5	25
Master	-	-	4	20
** *Sexual orientation* **				
Straight	19	63	na	na
Gay	6	33	na	na
Bisexual	5	17	na	na
** *Key population/most widely handled key population* **				
MSM (men who have sex with men)	11	37	7	35
PWID (people who inject drugs)	7	23	6	30
Female sex workers and their clients	3	10	3	15
Waria (trans women)	2	7	2	10
Others	7	23	2	10
** *HIV clinic location* **				
Central Jakarta	11	37	6	30
South Jakarta	3	10	3	15
East Jakarta	6	20	7	35
North Jakarta	6	20	1	5
Bekasi	4	13	-	-
Depok	-	-	1	5
Bogor	-	-	1	5
Tangerang	-	-	1	5
** *Region of origin* **				
Java	18	60	8	40
Sumatera	6	20	6	30
Jakarta	3	10	3	15
Sulawesi	2	7	2	10
Nusa Tenggara	1	3	-	-
Bali	-	-	1	5
** *Time since HIV diagnosis* **				
<10 years	19	63	na	na
>10 years	11	37	na	na
***ARV initiation*** (*months after diagnosis*)				
<1 month	9	30	na	na
<12 months	12	40	na	na
>12 months	9	30	na	na
** *Time on ARV* **				
<10 years	21	70	na	na
>10 years	9	30	na	na
** *ARV access/location of work* **				
Public hospital	20	67	7	35
Community health center (*Puskesmas*)	9	30	10	50
Private clinic	1	3	3	15
***Perceived health condition*** (*past 4 weeks*)				
Bad	2	7	na	na
Fair	4	13	na	na
Good	7	23	na	na
Very good	7	23	na	na
Extremely good	3	10	na	na
** *Profession/role in HIV care* **				
GP (general practitioner)	na	na	5	25
Nurse	na	na	5	25
Treatment companion/buddy/counselor	na	na	8	40
Psychologist	na	na	2	10
** *Professional experience in HIV care* **				
<7 years	na	na	9	45
>7 <14 years	na	na	11	55
** *Received training/workshop regarding HIV/ARV* **				
Yes	na	na	19	95
No	na	na	1	5

na = not assessed/not applicable.

**Table 2 tropicalmed-08-00138-t002:** Example of theme development.

Quotes		Category	Theme
PLHIV-OT	HSP		
*“We don’t want to think of it (ARV) as a drug, but consider it a vitamin. If we take vitamins, we want to be healthy, right?”*	*“I’ve always told PLHIV that this drug is a ‘vitamin’. Drug or medicine has such a bad stigma in their mind.”*	Lightening the “load” with euphemistic terminology	Intrapersonal level facilitators of treatment adherence at an advanced HIV age
*“I told my friends in the peer support group, that ARV is … beauty pills (laughing), because we can be beautiful and productive again with the pills.”*	*“They (PLHIV-OT) need to think of it as doing daily activities such as brushing their teeth every night, having dinner, or taking daily vitamin.”*		
*“My kids, my mom. My kids mostly. I want to be able to watch them grow up, finish college. That’s everyone hope I guess.”*	*“Like I’ve said before, the things that can help them (PLHIV-OT) to comply is their desire to live, their desire to be productive, and their desire to live with their family.”*	Meaning-making through goals and spirituality	
*“All I know that I’m committed to always pick up the ARV whenever it’s available for me as I still want to be alive.”*	*“When someone has a spirit of life, he must has a purpose in life. For example, he wants to have a family one day, or he wants to stay alive until the children get old. That’s a motivation.”*		

## Data Availability

Not applicable.

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
