# Peer review of "Barriers and Facilitators to HIV Treatment Adherence in Indonesia: Perspectives of People Living with HIV and HIV Service Providers"

_tropicalmed, 2023, doi:10.3390/tropicalmed8030138_

Round 1
Reviewer 1 Report
Il materiale è molto interessante e significativo. La lingua inglese è appropriata e comprensibile.
Author Response
Reviewer 1
- Il materiale è molto interessante e significativo. La lingua inglese è appropriata e comprensibile.
(Italian review: The material is very interesting and significant. English language is appropriate and comprehensible)
We sincerely appreciate your positive evaluation.
Reviewer 2 Report
Dear authors
I’ve carefully revised your manuscript “Barriers and Facilitators to HIV Treatment Adherence in Indonesia: Client and Healthcare Provider Perspectives”, this manuscript aims to explore the barriers and facilitators to Antiretroviral Therapy (ART) adherence.
Here my comments to further improve your work:
MAJOR
1. Please revise your abstract
2. Please revise your introduction, make sure to state the year and source for the HIV cascade estimates (estimated PLHIV in Indonesia are for which year?). Please update UNAIDS targets or justify why the country has not adopted the 95-95-95 UNAIDS targets.
3. Would recommend you use PLHIV, people living with AIDS is no longer recommended
4. Please make sure to revise the transition of your paragraphs and make sure to clearly state without being repetitive the objective of your study.
5. in the interview process, make sure to also mention if HSP were also conducted at a location chosen by the participant.
6. Not clear to me why table 1 is needed. I would suggest you revise it and include participant’s characteristics that would help to give a better context for the qualitative approach, if available, suggest you include some clinical characteristics as well, for example %who are virally suppressed, %with high levels of ART adherence, etc, %HSP trained on HIV adherence, on friendly service provision, etc. You can also include another table with the thematic analysis
7. Data should be presented logically, either in the order in which the methods were presented or according to the order that most effectively develops the hypothesis
8. Please further revise your conclusion for effectiveness and your recommendations should be based on your findings.
9. Please revise the strengths and limitations of your study.
10. Please consider revising your discussion and how you’re interpreting your results, considering the limitations and the strengths of your study.
11. Please make sure to discuss what are the implications of your research in clinical practice.
Minor comments
1. Please further revise your manuscript and the references for errors.
2. I would suggest you further revise the length of your manuscript; I would advise you to be more concise and to express ideas clearly and succinctly.
Author Response
Reviewer 2
I’ve carefully revised your manuscript “Barriers and Facilitators to HIV Treatment Adherence in Indonesia: Client and Healthcare Provider Perspectives”, this manuscript aims to explore the barriers and facilitators to Antiretroviral Therapy (ART) adherence.
Here my comments to further improve your work:
MAJOR
- Please revise your abstract
As per the reviewer’s request, we have revised our abstract. We hope it now better reflects the content of the paper.
Pre-revised version: HIV treatment adherence in Indonesia is still a major challenge. Previous studies have shown several barriers and facilitators to adherence. However, studies which provide a comprehensive analysis from both client and healthcare provider perspectives are limited, especially in Indonesia. We explored the barriers and facilitators to Antiretroviral Therapy (ART) adherence using a socioecological approach by interviewing 50 participants, comprised of 30 people living with HIV on treatment (PLWHA-OT) and 20 HIV service providers (HSP). Both groups of participants reported stigma as the major barrier at each level, starting from public stigma at the societal level, stigma in healthcare settings, until self-stigma at the intrapersonal level. Other barriers can be overcome by different facilitators at each level or can be the inverse of the barriers. For example, complicated bureaucracy at the health system level (i.e., HIV clinics) can be reduced with the help of buddies from NGOs. In the future, it is crucial to reduce stigma on the most abstract societal level, in order to enhance the facilitators at the subordinate levels and overcome the barriers and ultimately boost ART adherence.
Revised version: HIV treatment adherence in Indonesia is a major challenge. Although previous studies have demonstrated several barriers and facilitators to adherence, studies providing a comprehensive analysis from both PLHIV’ and HIV service providers’ perspectives are limited, especially in Indonesia. In this qualitative study with 30 people living with HIV on treatment (PLHIV-OT) and 20 HIV service providers (HSP), we explored, via online interviews, the barriers and facilitators to antiretroviral therapy (ART) adherence using a socioecological approach. Both PLHIV-OT and HSP reported stigma as a major barrier at each socioecological level, including public stigma at the societal level, stigma in healthcare settings, and self-stigma at the intrapersonal level. Stigma reduction must therefore be prioritized. PLHIV-OT and HSP also reported support from significant others and HSP as the foremost facilitator to ART adherence. The enablement of support networks is thus an important key to improved ART adherence. Overall, societal level and health system barriers to ART adherence should be addressed in order to remove barriers and enhance the facilitators at the subordinate socioecological levels.
- Please revise your introduction, make sure to state the year and source for the HIV cascade estimates (estimated PLHIV in Indonesia are for which year?). Please update UNAIDS targets or justify why the country has not adopted the 95-95-95 UNAIDS targets.
We have now revised the introduction using UNAIDS estimates from 2021. Also, we have explained that, although the governor of Jakarta signed the Paris Declaration in 2015, the 95-95-95 targets have not been achieved. Furthermore, we have worked on making the introduction more concise and more coherent.
Pre-revised version (line 28-32): However, adherence to antiretroviral therapy (ART) among PLWHA in Indonesia is low with poor rates of viral suppression. Of the estimated 610.000 PLWHA in Indonesia, only 54% are aware of their HIV status, 39% are on ART, and 30% have suppressed viral loads. These percentages are far below the 90-90-90 targets of UNAIDS.
Revised version (line 30-37): However, ART adherence among PLHIV in Indonesia is low. In terms of the 95-95-95 targets, the UNAIDS data for 2021 indicated that, of the estimated 610.000 PLHIV in Indonesia, only 66% are aware of their HIV status, 26% on ART, and no 2021 data was available on viral suppression. Clearly, Indonesia does not currently meet the 95-95-95 targets in spite of the governor of Jakarta having signed the Paris Declaration in 2015. This can occur due to stigma and discrimination, as well as criminalization of key populations (e.g., men who have sex with men or MSM, trans women) which make them reluctant to access HIV services.
- Would recommend you use PLHIV, people living with AIDS is no longer recommended
Thank you. We have used PLHIV and PLHIV-OT throughout the revised manuscript.
- Please make sure to revise the transition of your paragraphs and make sure to clearly state without being repetitive the objective of your study.
Thank you. We have thoroughly reviewed and revised the manuscript to ensure that transitions are present and the writing is coherent and concise. You will see this reflected throughout the manuscript.
- in the interview process, make sure to also mention if HSP were also conducted at a location chosen by the participant.
Thank you for pointing this out. We now mentioned that HSP also chose the location to be interviewed. This is reflected in line 140-142.
Pre-revised version (line 148-149): "The interviews were conducted in Bahasa Indonesia at a location chosen by the participant, usually at the participant’s house or bedroom, and no other people were present at the time of the interview."
Revised version (line 140-142): " Interviews were conducted in Bahasa Indonesia at a location chosen by the participants, usually the participant’s house or bedroom for PLHIV-OT or at the HIV clinic for the HSP. No other people were present at the time of the interview. "
- Not clear to me why table 1 is needed. I would suggest you revise it and include participant’s characteristics that would help to give a better context for the qualitative approach, if available, suggest you include some clinical characteristics as well, for example %who are virally suppressed, %with high levels of ART adherence, etc, %HSP trained on HIV adherence, on friendly service provision, etc. You can also include another table with the thematic analysis
We have now drafted a Table reflecting participants' characteristics (Table 1). In this table, we include age, gender, educational background, key population/most widely handled key population, area of HIV clinic, and region of origin for all participants. Also, for PLHIV-OT, we additionally present sexual orientation, time since diagnosis, time since ARV initiation, time on ARV, ARV access, and health status. For HSP, we also report location of work, profession/role in HIV care, professional experience in HIV care, and whether they ever received a training/workshop regarding HIV/ARV. Unfortunately, we do not have the requested clinical characteristics. We also drafted a table of example of theme development (Table 2).
- Data should be presented logically, either in the order in which the methods were presented or according to the order that most effectively develops the hypothesis
Thanks for pointing this out. In the revised manuscript, we present concepts in the same order throughout. We do this in the introduction (line 53-61) and then the result reflect this same order.
- Please further revise your conclusion for effectiveness and your recommendations should be based on your findings.
We have revised the conclusions by moving the last paragraph regarding theoretical and practical implications into the discussion, and make it more concise.
Pre-revised version: Stigma colors the barriers to adherence at every level. Besides stigma, there are other barriers which hinder the patient’s adherence starting from the treatment initiation until after years of treatment. Those barriers can be overcome by different facilitators at each level, can be the inverse of the barriers, or unique cultural-based facilitator. A society free of stigma and discrimination is certainly a hope for both PLWHA-OT and HSPs to achieve better adherence, but there were also resources identified can help to overcome certain barriers to adherence at every level. Nevertheless, it is crucial to intervene starting from the most abstract societal level, so that it will boost the facilitators at the subordinate levels.
This study provides a comprehensive explanation on the barriers and facilitators to ART adherence in Indonesia. We believe this study can add more knowledge on any studies regarding HIV treatment, particularly in Indonesia. In the future, we also hope that our findings can provide an input to make or adjust the policies for the treatment and welfare of PLWHA. As for the practical implications, both PLWHA-OT and HSPs can implement the facilitators mentioned in our findings to reduce the barriers to adherence. For example, the HSPs can suggest PLWHA-OT to address ARV euphemistically to minimize the tedious feeling in taking the medicine, or by helping the PLWHA-OT to set clear goals to increase their hope in continuing the treatment. Moreover, PLWHA-OT can proactively gain more HIV and ARV knowledge through reliable sources to help them having more understanding about the importance of the treatment, to try to maintain a good relationship with their providers, and, if possible, gain more support from their friends, family, or peer support group.
Revised version: Stigma is the foremost to ART adherence across every socioecological level. In addition, a number of other barriers can impede adherence starting from the treatment initiation until after years of treatment. Such barriers can be overcome by complementary facilitating factors that can ameliorate the barriers at each socioecological level. A society free of stigma and discrimination is certainly a hope for both PLHIV-OT and HSP not only to achieve better ART adherence, but also improve quality of life for PLHIV. In that context, support from significant others and HSP remains imperative. Overall, it is crucial to intervene starting from the most abstract societal level, so that it facilitators can be bolster and barriers can be overcome at the subordinate levels.
- Please revise the strengths and limitations of your study.
As per the reviewer's request, we have revised the limitations. We now discuss the possible impacts of sampling approach, both in terms of location and in terms of sampling approach, as well as the fact that PLHIV participants were already on ART and that this study occurred during the Covid-19 pandemic. This is reflected in lines 927 to 953.
- Please consider revising your discussion and how you’re interpreting your results, considering the limitations and the strengths of your study.
We have revised the discussion by focusing on stigma as the major barrier at every level, adding several ways to reduce stigma, adding the clinical, theoretical, and practical implications. We also added the convenience sampling as the limitation and stated that the generalizability is not the goal of qualitative research, also the fact that PLHIV-OT who are already in treatment for years may have more favorable views on ART as the limitation.
- Please make sure to discuss what are the implications of your research in clinical practice.
Thank you for this. We now discuss clinical implications of our findings, namely that HCP provide a reliable and easy to access, one-stop treatment service for PLHIV-OT with consistent long term follow-up, that HCP create optimal conditions for the establishment and maintenance of a solid support system for PLHIV-OT either from their significant others or peer support group, and that HSP continuously improve their ability and knowledge about HIV care and counseling, potentially through training sessions and workshops, so that they can provide the most optimal service. This can be found on lines 917-926.
Minor comments
- Please further revise your manuscript and the references for errors.
We have fully reviewed and revised the manuscript to improve coherence and conciseness. We have also done our best to remove any grammatical and typing errors. Furthermore, we have fixed the references.
- I would suggest you further revise the length of your manuscript; I would advise you to be more concise and to express ideas clearly and succinctly.
Indeed, this paper needed a thorough read through to tighten up the text and improve coherence. We have now this and hope that you agree that the revised manuscript is a substantial improvement on the original manuscript.
Reviewer 3 Report
Hutahaean et al aimed to study factors associated with non-adherence to HIV therapy in Jakarta, Indonesia, where overall rates of viral suppression in the HIV population are low. They used a socioecological approach using semi-structured interviews of both patients and providers. They found several themes associated with non-adherence. Personal, societal, and institutional stigma, inadequate healthcare system coverage, side effects, mental health, and bureaucratic hurdles including access to ARVs were given as contributors to non-adherence. Family support, non-judgmental providers, spirituality, and support groups were noted as contributors to adherence. Interestingly, ‘fear appeal’ was also noted as a motivator to stay adherent, and social media was mentioned as both a positive and negative influencer. They provide potential solutions to improving adherence based on their findings.
The manuscript is well organized. The methods section is detailed and is easy to follow. Overall, the study provides important data to inform any future Indonesian public health policies that are aimed at improving adherence to HIV medications. My general comments are below, followed by more specific comments:
My main critique of the study has to do with the method for selection of the respondents. The providers were selected via purposive and snowball sampling, which does not provide a good representation of the providers in the community. Providers referred by other providers may be more likely to have similar views. This can skew the results and does not provide good cross-sectional representation of provider views on non-adherence. The authors should discuss their rationale for selecting this sampling method and include it in the limitations section.
It is also not clear how the 30 PLWHA were selected – were they also selected via purposive and snowball sampling? Please make this clear in the methods. If they also had the same sampling methods, then as above, provide the rationale and including as a limitation.
In the limitations section, the authors point out that some participants reflect the diversity of the Indonesian population but suggest that future studies select patients from different provinces. However, the manuscript does not describe the hometowns of the participants. If this is an important limitation, the manuscript should describe the demographics of the participants in more detail. Lines 115-129 are dedicated to describing the participants. However, adding a table with more detail about the participants would be helpful. The table should include geographic area as well as the specific jobs of the HIV service providers.
Below are more specific comments:
Lines 17-18: Other barriers can be over-come by different facilitators at each level or can be the inverse of the barriers. I am not sure what this sentence means – what is the inverse of the barriers? Please clarify.
Lines 18-21: For example, complicated bureaucracy at the health system level (i.e., HIV clinics) can be reduced with the help of buddies from NGOs. In the future, it is crucial to reduce stigma on the most abstract societal level, in order to enhance the facilitators at the subordinate levels and overcome the barriers and ultimately boost ART adherence.
Again, this is confusing – what do the authors mean by ‘buddies’? This is a very vague conclusion – please be more concrete than ‘it is crucial to reduce stigma’.
Line 40: This is incorrect – while good adherence often leads to undetectable virus, this is not the definition. I think it would be more accurate to say that good adherence is associated with viral suppression, which decreases transmission.
Line 170: Who translated the other PLWHA interviews?
Line 323: I think there’s a typo – is the ‘and’ supposed to be ‘an’?
Line 875: Change ‘live’ to ‘life’
I thank the editors for the opportunity to review this manuscript and I applaud the authors for their efforts on this interesting and valuable study.
Author Response
Reviewer 3
Response:
Dear Reviewer 3,
We thank you for your opinion that our study is interesting and valuable. We have revised the manuscript according to your reviews and we hope you agree with the revision.
Hutahaean et al aimed to study factors associated with non-adherence to HIV therapy in Jakarta, Indonesia, where overall rates of viral suppression in the HIV population are low. They used a socioecological approach using semi-structured interviews of both patients and providers. They found several themes associated with non-adherence. Personal, societal, and institutional stigma, inadequate healthcare system coverage, side effects, mental health, and bureaucratic hurdles including access to ARVs were given as contributors to non-adherence. Family support, non-judgmental providers, spirituality, and support groups were noted as contributors to adherence. Interestingly, ‘fear appeal’ was also noted as a motivator to stay adherent, and social media was mentioned as both a positive and negative influencer. They provide potential solutions to improving adherence based on their findings.
The manuscript is well organized. The methods section is detailed and is easy to follow. Overall, the study provides important data to inform any future Indonesian public health policies that are aimed at improving adherence to HIV medications. My general comments are below, followed by more specific comments:
Response: Thank you for the compliment and for your guidance in improving this manuscript.
My main critique of the study has to do with the method for selection of the respondents. The providers were selected via purposive and snowball sampling, which does not provide a good representation of the providers in the community. Providers referred by other providers may be more likely to have similar views. This can skew the results and does not provide good cross-sectional representation of provider views on non-adherence. The authors should discuss their rationale for selecting this sampling method and include it in the limitations section.
Response: In qualitative research, convenience and snowball sampling is common and accepted. Generalizability is not the aim (see e.g. Stutterheim & Ratcliffe, 2021). Rather, the aim is gain a contextualized understanding of complex phenomenon, as is the case for adherence to ARV in Indonesia, something that has not extensively been studied. Nonetheless, it is advisable to indeed seek diversity and representation of multiple views in qualitative research. For this reason, we triangulated data across both health service providers and PLHIV, and sought diverse representation across people of different genders, sexual orientation, hometown, and, specifically for health service providers, professional backgrounds.
It is also not clear how the 30 PLWHA were selected – were they also selected via purposive and snowball sampling? Please make this clear in the methods. If they also had the same sampling methods, then as above, provide the rationale and including as a limitation.
Response: A convenience sample PLHIV were recruited purposively and then through snowball sampling. This is now better described in methods section in lines 100-102, and discussed further in the limitations section (lines 942-948)
In the limitations section, the authors point out that some participants reflect the diversity of the Indonesian population but suggest that future studies select patients from different provinces. However, the manuscript does not describe the hometowns of the participants. If this is an important limitation, the manuscript should describe the demographics of the participants in more detail. Lines 115-129 are dedicated to describing the participants. However, adding a table with more detail about the participants would be helpful. The table should include geographic area as well as the specific jobs of the HIV service providers.
Response: Thank you for your suggestion. In our revised manuscript, we added Table 1, which now reflects a range of participant characteristics, including the PLHIV-OT's hometown/region of origin and HSP's occupations. Also, in the limitations, we discuss the possible implications of recruiting in and around Jakarta. This is detailed in lines 937-942.
Below are more specific comments:
- Lines 17-18: Other barriers can be over-come by different facilitators at each level or can be the inverse of the barriers. I am not sure what this sentence means – what is the inverse of the barriers? Please clarify.
We agree that this text was not particularly clear. We have therefore revised the abstract and the discussion where this claim was made so that it is clearer. This can be found in lines 18 – 23.
- Lines 18-21: For example, complicated bureaucracy at the health system level (i.e., HIV clinics) can be reduced with the help of buddies from NGOs. In the future, it is crucial to reduce stigma on the most abstract societal level, in order to enhance the facilitators at the subordinate levels and overcome the barriers and ultimately boost ART adherence. Again, this is confusing – what do the authors mean by ‘buddies’? This is a very vague conclusion – please be more concrete than ‘it is crucial to reduce stigma’.
We agree that the word ‘buddies’ and the text was confusing. We have revised the abstract which can be found in lines 11-23.
- Line 40: This is incorrect – while good adherence often leads to undetectable virus, this is not the definition. I think it would be more accurate to say that good adherence is associated with viral suppression, which decreases transmission.
Thank you for pointing this out. We agree that good adherence can lead to undetectable viral load, but it doesn't mean that all PLHIV with a good adherence are undetectable. We have changed the sentence in this line as suggested, emphasizing the good adherence is associated with viral suppression.
- Line 170: Who translated the other PLWHA interviews?
The other PLHIV transcripts were not translated into English. The first author coded the rest of the PLHIV-OT transcript’s in Bahasa Indonesia because we found too much typical phrases and informal words in Bahasa Indonesia which had (almost) no equivalent words in English. Therefore, we decided to code it in its original language (Bahasa Indonesia) to minimize the misinterpretation.
- Line 323: I think there’s a typo – is the ‘and’ supposed to be ‘an’?
Thank you for your correction. It is a typo and it supposed to be 'an'. We have changed it according to your feedback. Also, as you can see, we have fully revised the manuscript for coherence and conciseness, and have done our best to remove grammatical and typing errors.
- Line 875: Change ‘live’ to ‘life’
Thank you for your suggestion. The second 'live' should be 'life'. We have changed it according to your feedback.
Round 2
Reviewer 2 Report
Dear authors, thanks for considering my comments to improve your work.
Additional suggestions:
1. Please note the difference between country's progress towards UNAIDS targets and the adoption of the targets. My previous suggestion was to provide updated data based on 95-95-95 targets or justify why the country has not adopted these new targets.
2. For participant characteristic, avoid being duplicative, summarize key information and refer readers to the table.
3. Please synthesize the explanation about why interviews where conducted via phone/WhatsApp...
4. Suggest to further reduce result section, I still found it to be lengthy
5. Revise your discussion and conclusion for effectiveness, I still found it to be lengthy
Author Response
Referee
Dear authors, thanks for considering my comments to improve your work.
Additional suggestions:
1. Please note the difference between country's progress towards UNAIDS targets and the adoption of the targets. My previous suggestion was to provide updated data based on 95-95-95 targets or justify why the country has not adopted these new targets.
We have provided the current data based on 95-95-95 targets and several reasons why these targets have not been achieved. The most recent data on 95-95-95 targets does not contain information on the last column, but data from previous years showed poor performance in terms of viral suppression. We have added those references and also extended the reasons provided in the literature, ranging from poor retention in care over legal barriers to psycho-social determinants. The revised paragraph now reads:
In terms of the 95-95-95 targets, the UNAIDS data for 2021 indicated that, of the estimated 610.000 PLHIV in Indonesia, only 66% are aware of their HIV status, 26% on ART, and no 2021 data was available on viral suppression [4], but earlier data also showed a very poor rates of viral suppression [5,6]. Clearly, Indonesia as a whole does not meet the 95-95-95 targets [4] despite of efforts, such as the 2012 continuum of care government initiative (Layanan Komprehensif Berkesinambungan) [7] and regional activities, such as the governor of Jakarta having signed the Paris Declaration in 2015 [8]. A number of reasons have been brought forward, such as poor retention in care [2], but also stigma and discrimination in society, as well as criminalization of key populations (e.g., men who have sex with men or MSM, trans women) which creates barriers for accessing HIV services [9–13].
For participant characteristic, avoid being duplicative, summarize key information and refer readers to the table.
We have reduced the amount of information provided for the participant characteristics and refer the readers to Table 1 for a more detailed presentation of sample characteristics. The revised paragraph now reads:
PLHIV-OT’s mean age was 38.7 years with the average of 6.7 years on ARV treat-ment. Among them, more than half were cisgender men and self-identified as straight. The HIV service providers had a mean age of 34.7 years with more than half of them had a professional experience in HIV care for 7 to 14 years. In terms of formal educational attainment, 35% of the HSPs had vocational training and 25% had a bachelor’s degree with only one of them never received any HIV-related formal training. Additional details are depicted in Table 1.
Please synthesize the explanation about why interviews were conducted via phone/WhatsApp...
We have extended the section explaining the reasons for conducting the interview via phone/WhatsApp calls, that is to minimize direct contact with the participants and to keep them safe from any COVID-19 risks. The revised paragraph now reads:
Pre-revised version: We utilized online mobile applications such as WhatsApp audio or video call, or online chat and video telephony software platforms such as Zoom.
Revised version: We utilized online mobile applications such as WhatsApp audio or video call, or online chat and video telephony software platforms such as Zoom to avoid direct contact with participants and ensure their safety from any COVID-19 risks.
4. Suggest to further reduce result section, I still found it to be lengthy
We have cut the result section overall by trimming verbiage and making it more concise, deleting certain quotes and presenting the most relevant quotes in each theme, while at the same time avoiding duplicate results presentation if similar results occurred at several levels of analysis. We hope that the current presentation is more flashed out and more accessible to the reader.
5. Revise your discussion and conclusion for effectiveness, I still found it to be lengthy
Given the changes in the Result section, we have revised the Discussion and Conclusion as well. We have removed sidelines in the argumentation and focused on the core results. Overall, the number of pages has dropped from 28 (before minor revision) to 25.
Round 3
Reviewer 2 Report
no further comments, thanks for considering my suggestions